# Mitigating Breast-Cancer-Related Lymphedema—A Calgary Program for Immediate Lymphatic Reconstruction (ILR)

**Melina Deban** [1,*]**, J. Gregory McKinnon** [1] **and Claire Temple-Oberle** [1,2]

1    Division of Surgical Oncology, Tom Baker Cancer Center, Calgary, AB T2N 4N2, Canada
2    Division of Plastic Surgery, Tom Baker Cancer Center, Calgary, AB T2N 4N2, Canada
*    Correspondence: melina.deban@ucalgary.ca

**Abstract:** With increasing breast cancer survival rates, one of our contemporary challenges is to improve the quality of life of survivors. Lymphedema affects quality of life on physical, psychological, social and economic levels; however, prevention of lymphedema lags behind the progress seen in other areas of survivorship such as breast reconstruction and fertility preservation. Immediate lymphatic reconstruction (ILR) is a proactive approach to try to prevent lymphedema. We describe in this article essential aspects of the elaboration of an ILR program. The Calgary experience is reviewed with specific focus on team building, technique, operating room logistics and patient follow-up, all viewed through research and education lenses.

**Keywords:** lymphedema; immediate lymphatic reconstruction; breast cancer

## 1. Introduction and Rationale for Performing Immediate Lymphatic Reconstruction

### 1.1. Spotlight on Survivorship and the Burden of Lymphedema

Survival rates for breast cancer have markedly improved over the last few decades. The Canadian Cancer Society reports 5-year survival rates of 100% for Stages 0 and 1, 93% for Stage 2 and 72% for Stage 3 [1]. Surgical, locoregional and systemic therapies have decreased in intensity and become more targeted [2,3]. One of our contemporary challenges is to improve the quality of life for survivors' experiences. To this end, breast reconstruction is offered widely to appropriate candidates, and often immediately at the time of mastectomy [4]. In contrast, lymphedema has been accepted as a sequela of therapy, with many women experiencing a high risk of long-term upper limb lymphedema [5]. Lymphedema affects quality of life on physical, psychological, social and economic levels [6–8]. Within the realm of survivorship, important work has been conducted in fertility preservation [9], minimizing chronic chemotherapy toxicities [10] and minimizing the impact of surgery [4,11]. Though techniques have been developed to try to improve lymphatic flow, both with supportive therapy and surgery [12], there has been less focus on the prevention of lymphedema. A proactive approach such as immediate lymphatic reconstruction might be the answer.

### 1.2. Surgical Options for Lymphedema Prevention

Lymphedema is thought to result from the disruption of the lymphatic channels draining the upper extremity during axillary lymph node dissection (ALND). Work leading to the development of ILR includes axillary reverse mapping (ARM) [13–15]. ARM allows for the identification of the lymphatic channels draining the arm using either blue dye [14], technetium sulphur colloid [15] or both [13], which can in turn be spared during dissection to prevent lymphedema. Boccardo and colleagues described a microsurgical technique for immediate lymphatic reconstruction (ILR) [16,17]. Identification of lymphatics draining the upper extremity can be anastomosed to branches of the axillary vein at the time of ALND. It was hypothesized that restoring lymphatic drainage of the arm to the axillary vein could prevent lymphedema [16].

### 1.3. What Is the Current Place of Axillary Lymph Node Dissection?

An alternative prevention strategy to lymphedema is avoiding ALND altogether, and asking ourselves whether we should be performing ALND at all. Two landmark studies have revolutionized the landscape of lymph node dissection in breast cancer: ACOSOG Z0011 [18] and AMAROS [19]. ACOSOG Z0011 showed that it was safe to avoid ALND in patients having a lumpectomy for cT1-2N0 breast cancer with 1–2 positive axillary lymph nodes. AMAROS demonstrated equivalent locoregional control for patients undergoing axillary radiation therapy versus ALND. Lymphedema rates favoured observation in ACOSOG Z0011 and radiation therapy in AMAROS when compared to ALND.

### 1.4. Neoadjuvant Therapy and the Axilla—What to Do?

A question still arises in patients undergoing neoadjuvant therapy with a complete clinical axillary response. There is increasing evidence for the feasibility of sentinel lymph node biopsy (SLNB) without completion of ALND. ACOSOG Z1071 was the first trial to show that the false-negative rate after neoadjuvant chemotherapy (NAC) could be reduced to 10.8% with the use of the dual-tracer technique [20], and to 6.8% if the clipped node was removed [21]. Similarly, SENTINA [22], SN FNAC [23] and Caudle and colleagues [24] demonstrated that FNR was below 10% if two or more lymph nodes were retrieved, the dual-tracer technique was employed and/or the clipped note was retrieved. Recently, a nuanced approach to the axilla has been proposed, with a shared decision-making patient care pathway [25].

However, the number of patients with upfront cT4 or cN2/3 disease (prior to NAC) was very low in all of the aforementioned studies [26]. It is unclear whether these patients should systematically undergo ALND after NAC. Moreover, although many studies have examined the false negative rate post-NAC, few studies have examined the oncological outcomes for patients having SLNB after NAC [27]. Lastly, rates of response to neoadjuvant therapy remain limited for subtypes such as estrogen and progesterone (ER/PR)-positive breast cancer, in the order of 15–20% [28,29]. It seems that even if we are targeting patients more specifically for ALND, and overall numbers of patients undergoing dissection are decreasing, ALND in highly selected patients is here to stay with the current body of knowledge.

### 1.5. Maybe This Is Where ILR Finds a Home

In this context, ILR is appealing as a prevention strategy. There is growing evidence to support its use in breast and other cancers, with a recent meta-analysis showing a risk ratio of 0.22 for lymphedema with ILR, yielding a number needed-to-treat of four [30]. Scaglioni and colleagues performed a systematic review of 939 patients in which 50–100% reported subjective symptom relief. All 18 studies had an objective decrease in circumferential measurements [31]. Jorgensen and colleagues' study found a relative risk ratio of 0.33 for prophylactic ILR when compared to no-prophylactic treatment [32]. A retrospective review of 32 patients having successfully undergone ILR after ALND showed an impressive lymphedema rate of only 3.1% [33]. Similarly, after 4 years of follow-up, Boccardo and colleagues found that lymphedema occurred in 3 patients out of 74 having undergone ALND and ILR for breast cancer [17].

### 1.6. Why Forming a Team and Getting Ready to Perform ILR Makes Sense

We are awaiting results from six RCTs investigating the effectiveness of ILR in the prevention of lymphedema (NCT03428581, NCT03941756, NCT04241341, NCT04328610, NCT05366699 and NCT05136079). The six research groups include the Mayo Clinic, MD Anderson Cancer Center, Memorial Sloane Kettering Center, Pusan National University Hospital, Stanford University and the University of Calgary. Given the promising results from non-randomized trials, in the meantime, surgeons should seize the opportunity to learn and perfect this technique. We describe in this article essential aspects of the elaboration of an ILR program.

## 2. Initial Steps

### 2.1. First Thing's First—Fostering the Team

The cornerstone of implementing a new program lies in the attitude of the team. The elements that are key to success are practitioners who are interested, motivated and unhindered by early adoption who have the necessary skillsets, a collaborative, academic environment and a supportive institution. Because both the resecting and reconstructing surgeons are present throughout the entire procedure, a significant time and resource commitment which proves beneficial in the long run, the relationships have to be amiable and positive (Figure 1).

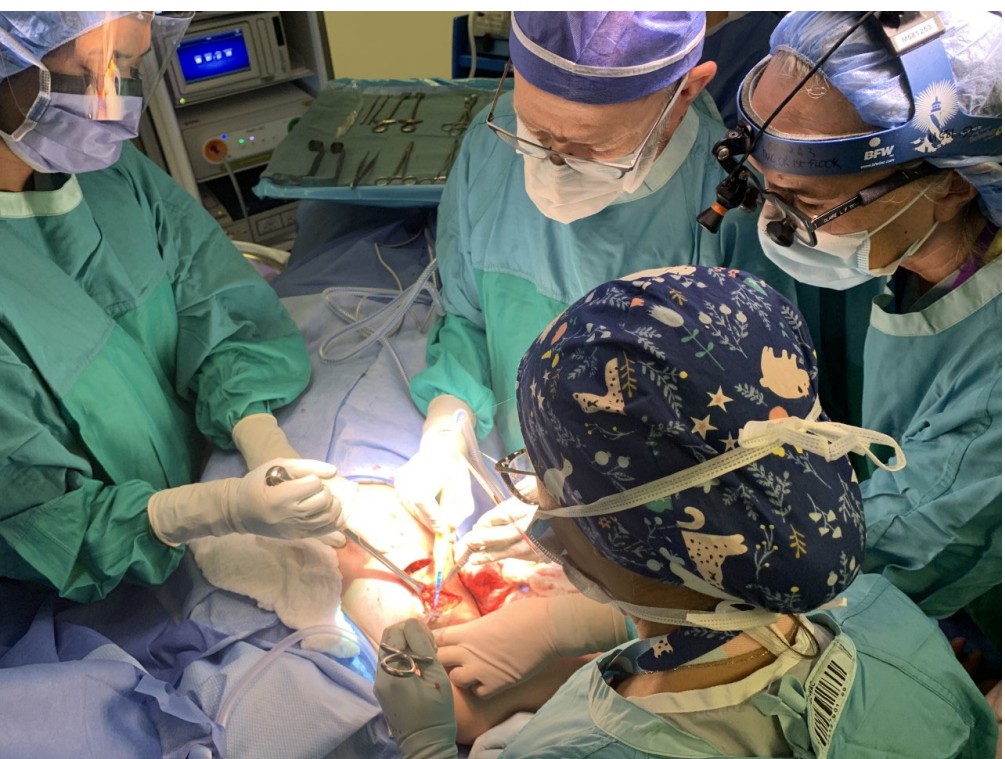

**Figure 1.** The team set up at an immediate lymphatic reconstruction in Calgary.

Fellowship microsurgical training for the plastic surgeon is essential to be able to perform this technique. Surgical oncologic training is an asset for the resecting surgeon. Cross training in terms of electives in plastics or surgical oncology can be helpful at the fellowship level if available. A local simulation lab, such as the Advanced Technical Skills Simulation Laboratory (ATSSL) in Calgary [34], provides invaluable opportunities to practice the intraoperative set up and the switching up of the main operators, and for microsurgeons to practice supermicrosurgical anastomoses as well as for the surgical oncologist to become facile using the microscope and microinstruments in order to be the first assist for the lymphatic reconstruction when necessary.

The importance of skilled microsurgical and breast oncological operating room nursing support cannot be understated. The implementation of an ILR program is resource-intensive for everyone involved at first but, in our experience, it has been well received. Rather than pushback, the most common remark from ancillary staff has been why this procedure was not instigated years ago.

## 2.2. Selecting the Patients

Indications for ILR remain broad in our institution. ILR is offered to all patients undergoing upfront ALND. For patients undergoing neoadjuvant chemotherapy, a patient care pathway is developed [25]. Following neoadjuvant chemotherapy, patients are reassessed via clinical exam and ultrasound. If persistent or suspicious nodes are found, they undergo ALND and ILR is offered. Similarly, if there are contraindications to sentinel lymph node biopsy such as multiple gross/matted nodes or inflammatory breast cancer, they are offered ALND with concurrent ILR. There are no frank contraindications in attempting ILR. Prior or future radiotherapy is not considered in patient suitability for ILR.

Preoperative measurements are taken at the preoperative visit with the plastic surgeon. Limb measurement is described in more detail in Section 3.1. No specific preoperative therapy or measures need be taken to optimize the procedure.

## 2.3. The Technique

### 2.3.1. Part 1: Macrosurgery

The procedure begins with an injection of patent blue dye in the upper inner arm and ICG dye in the hand. Lymphatics are mapped up to the axilla with the assistance of the handheld SPY-PHI (Stryker, Kalamazoo, MI, USA) device. Blue dye and ICG are augmented as necessary from 0.5 to 1 mL increments. The surgical oncologist/general surgeon begins the node dissection but leaves the lateral aspect until last. Throughout the dissection, the plastic surgeon and surgical oncologist often swap positions to look for lymphatics, particularly superolaterally in the axilla but also to assist with vein dissection and protection to preserve length. The lymphatics can be quite superficial and can be prematurely shortened on the way in, which can preclude successful anastomosis. Thus, the plastic surgeon must map the entry point of lymphatics into the axilla and be present at the initial incision to be sure the lymphatics are not cut early, before they are dissected toward the nodal specimen. Lowering the axillary incision to just below the inferior aspect of the hair-bearing axilla can permit gaining length on superficial lymphatics by dissecting them from the axillary skin flap if needed, toward the arm. Great care is taken not to injure the thoracodorsal artery, vein and nerve, as well as the long thoracic nerve. When possible, a branch of the axillary vein is kept intact for the anastomosis. If unavailable, the thoracodorsal vein can be used, although if used for an end to end anastomosis, the patient could not have a future latissimus flap for future issues such as chest wall coverage or breast reconstruction. Anywhere between one to five lymphatics of suitable calibre are identified, ideally just under a millimeter in size.

We have learned to avoid placing microclamps on candidate lymphatics during the dissection as they have little purchase and are often dislodged during axillary dissection. We use the Dr. Evan Jost suture/clip technique (Figure 2), named after a surgical oncologist who developed this idea in our melanoma ILR procedures. It is a natural example of the cross pollination of ideas which occurs between specialists, and something your teams will find as well. Microsurgeons are used to putting clamps on vessels for later anastomosis. The suture/clip technique involves affixing a 5 mm tag of 3-0 vicyl with a microclip at the furthest end of the lymphatic, which stands up well to sponging and traction during the axillary dissection.

### 2.3.2. Part 2: Microsurgery

Once the specimen is removed, hemostasis is ensured at a hypertensive blood pressure with a Valsalva maneuver or with a brief dose of vasopressor. A postoperative hematoma is problematic and risks avulsing the anastomosis either through distension of the tissues or operative re-exploration. Regarding exposure, the placement of deep self-retaining retractors makes the microsurgery difficult, as they create a deeper hole and push the tagged arm lymphatics laterally. It is easier to invert the medial skin, suturing it right down to the lateral chest wall (minding the long thoracic nerve), and then placing silk sutures on the lateral aspect of the incision to lightly retract it. Firm retraction distorts the arm

lymphatics and pulls them away from their target vein. Here is where the lymphatics can be effectively elongated by dissection toward the arm, if the initial axillary incision had been lowered. A 10 × 20 mm blue background is placed on a platform formed by a sponge gently pushed into the depths of the axilla.

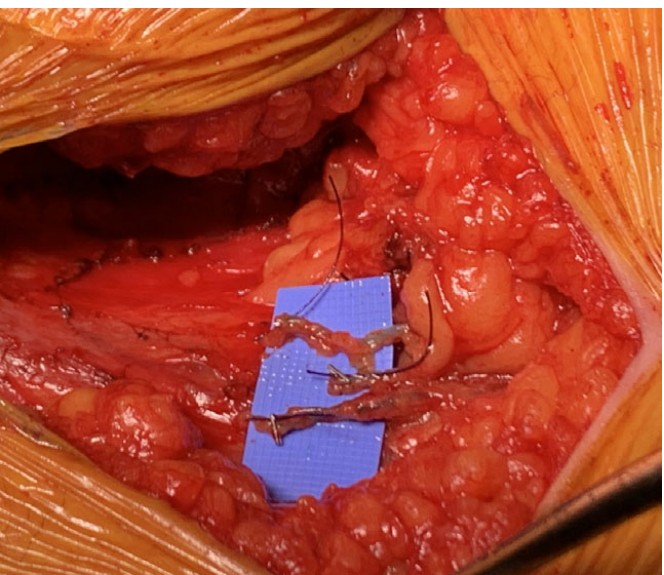

**Figure 2.** Suture/clip technique to identify lymphatics.

The operating microscope is brought in and set up with the patient in the Trendelenburg position and with the operating axilla up, in airplane. The main operator sits in the axillary position and the assistant stands above the table with an arm on either side of the patient's head, which must be carefully padded and eye-protected at the beginning of the case. Using the microdissection technique, the lymphatics and the vein are prepared. A microclamp is now placed on the lymphatic to prevent dye spillage when the suture/clip is cut off. However, do not remove the suture/clip until later, as having the lymphatic clipped makes it distend and prevents the microneedle from inadvertently passing through the lymphatic which would essentially ligate it.

A single- or double-approximating clamp is placed on the vein, depending on whether an end to end or an end to side technique is used, respectively. An end to end intussusception technique is our commonest technique in the axilla (in contradistinction to an end to side technique used in inguinal ILR). The vein should be transected as distal as possible, and be 5 mm or more distal to a valve, to prevent back pressure into the lymphatic. The lymphatic is secured with a 10-0 horizontal mattress, passed first through the vein, then to the outermost layer of the lymphatic just before the suture/clip is snipped off and finally back out through the vein again, drawing the open lymphatic deeply into the vein. This is augmented with one or two peripheral sutures from the vein end, barely biting the external surface of the lymphatic. This is carried out sequentially on all lymphatics. If there are still vacant areas of the vein lumen not receiving lymphatics, a 9-0 nylon simple suture can be placed to prevent lymphatic leakage. There should not be back bleeding if the repair has been made with a valve immediately upstream. All venous and lymphatic clamps are released to ensure no leakage. A proximal strip test shows blue dye visually and ICG dye (with the SPY) progressing proximally into the vein (Figure 3). Though use of lymphoscintigraphy has also been described [16] to verify the patency of the anastomosis, it is not routinely used at our institution in the intraoperative or postoperative setting. However, if the patient develops lymphedema, surgical and non-surgical options are contemplated for management, and lymphoscintigraphy and SPY imaging assist in decision making for the next steps [35].

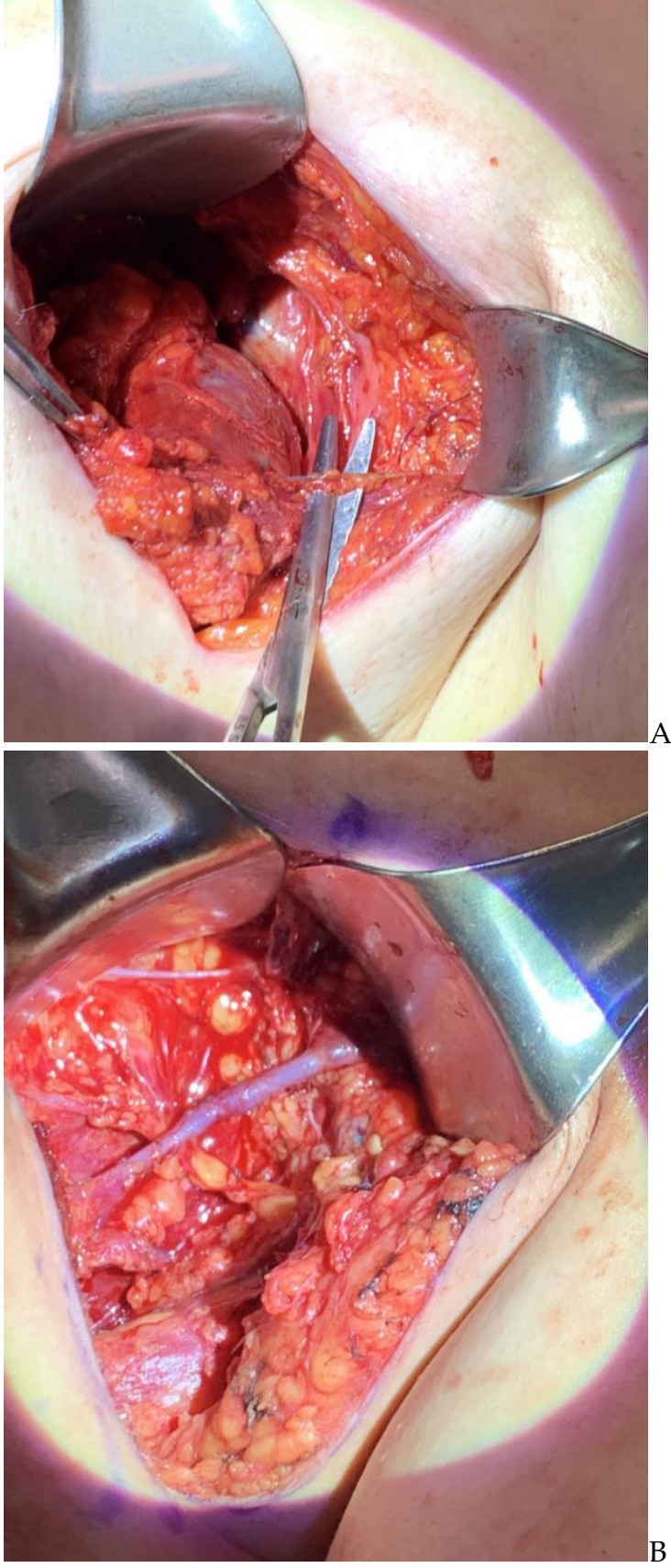

**Figure 3.** *Cont.*

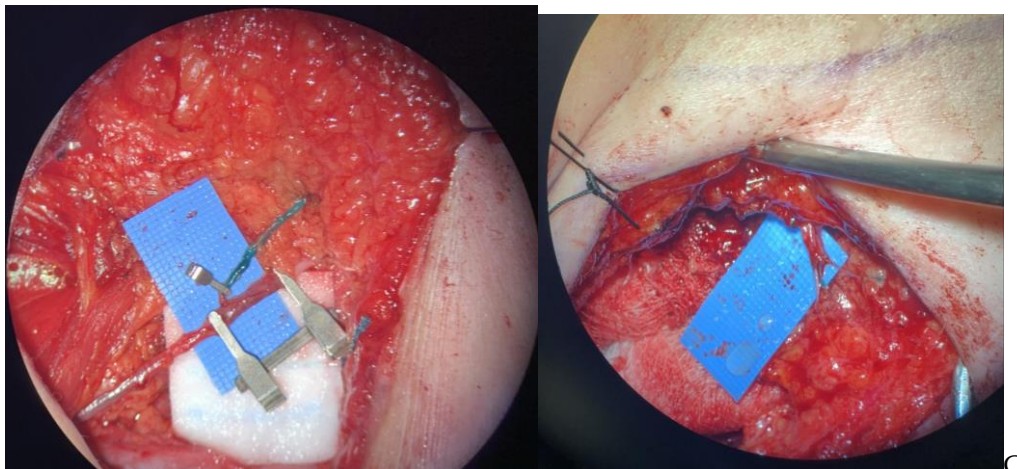

**Figure 3.** (**A**): Identification of a lymphatic channel. (**B**): Identification of suitable vein. (**C**): Preparing for an end to side lymphaticovenous anastomosis; picture on right shows multiple tiny intussuscepted lymphatics end to end into a recipient vein.

The arm can be taken through a range of motion to ensure no tension on the repair. If the repair is tight with the arm overhead, the patient is counselled postoperatively to limit abducting the shoulder beyond 90 degrees for two weeks, though we have found this to be generally unnecessary. A surgical drain is sutured externally to prevent slippage, and internally the drain is sutured with a 4-0 chromic to the chest wall, away from the anastomosis. Closure is carried out as per the usual fashion, being gentle with traction on the lateral flap to avoid avulsing the repair.

### 2.4. Learning the Intraoperative Dance

Team dynamics are essential to the technique. The general surgeon/surgical oncologist and the plastic surgeon frequently pause and swap operative and assistant roles in order to properly identify lymphatics but also to avoid compromising the oncologic nature of the surgery. Both surgeons are present from the beginning of the case to perform the injection of the tracers and the plastic surgeon typically monitors the proximal march of the dye in the lymphatics. They identify the lymphatic channels early on in the surgery and dissect them carefully to preserve maximal length. The general surgeon/surgical oncologist takes great care not to sever those lymphatics short, but also to preserve smaller calibre veins that usually would generally be divided in the ALND specimen. Without outflow opportunities, the lymphatic reconstructive procedure cannot proceed [16,17,33,36–38].

### 2.5. All the Extra Stuff

Super microsurgical equipment, 9-0, 10-0 and 11-0 sutures and a microscope with high magnification are the basic requirements. While indocyanine green (ICG) technology and patent blue dye are an ideal combination to map lymphatics, blue dye is ubiquitous and can be used to perform ILR before programs have ICG. ICG has the advantage of being easily traced with a scanner along the length of the lymphatic channel up the limb, while blue is more easily visible in the surgical bed, alerting the general or plastic surgeon of the proximity of a lymphatic needing preservation. If a lymphatic channel containing ICG is severed, ICG spill can compromise identification of other channels in the surgical bed if only ICG is used.

### 2.6. Operating Room Size and Time

In terms of operating room logistics, large operating rooms allowing space for a microscope, ICG/SPY equipment and a large team are preferred. The additional time added to a regular ALND procedure, especially at the start of one's learning curve, is largely rewarded by potential avoidance of lymphedema and by the patient's improved quality of life. The additional time taken at the beginning of the learning curve is generally reduced to 30 min once the two surgeons and nursing staff are familiar with each part of the procedure. Of note, ILR's cost effectiveness has been studied with encouraging results [39], with an incremental cost utility ratio of USD 1587.73 per quality-adjusted life year.

### 2.7. Important for Patients to Know

Patients typically consent to the procedure knowing that there will be permanent tattooing where the patent blue is injected. We have moved the injection from where we initially injected it in the volar wrist crease (Figure 4) to the upper inner arm just proximal to the elbow, to hide the tattoo better. The tattoo fades dramatically over time, but generally a remnant persists. In our experience, no patient has declined ILR based on this reason. As mentioned in Section 2.3, postoperatively, patients may be instructed not to abduct the shoulder beyond 90 degrees for two weeks, though this is not necessary in our experience.

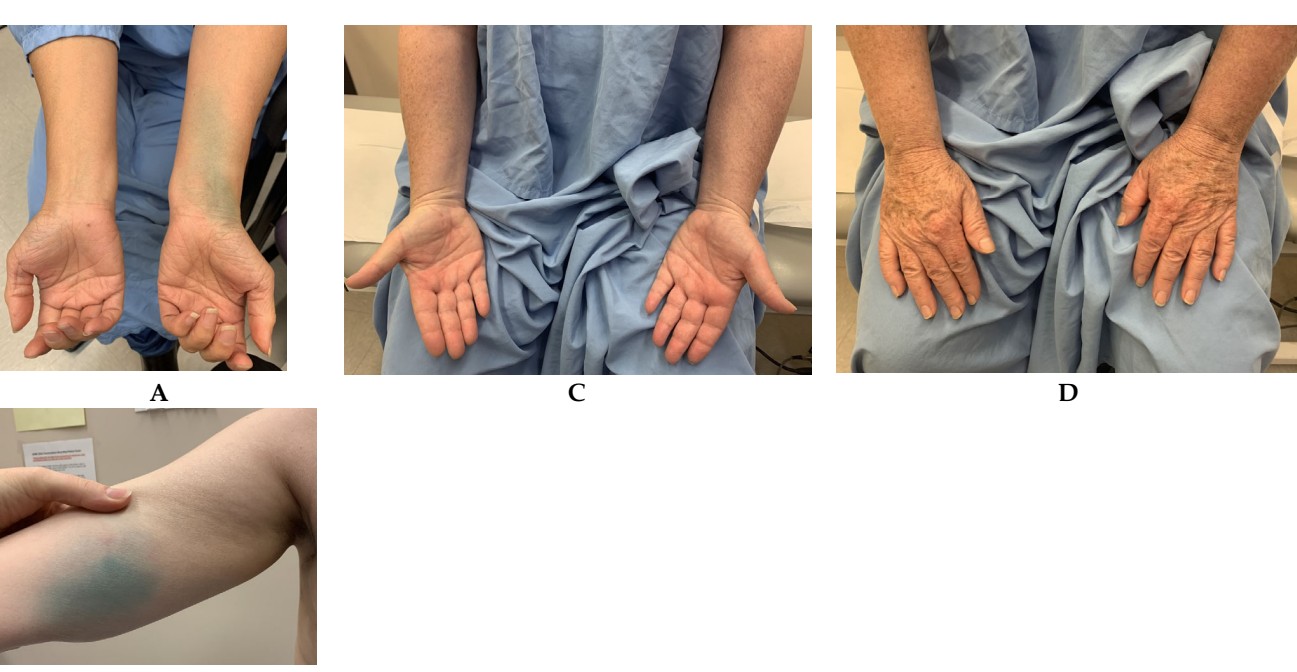

**Figure 4. (A):** Initially, we placed the blue dye injection on the volar wrist. (**B**): Now, we inject the upper inner arm for better camouflage. (**C,D**): Volar and dorsal photos demonstrated no visible lymphedema. Comparative limb volumes in right to left arms are similar.

### 2.8. Who Else Needs to Be on the Team?

Postoperatively, it is important to be linked to a cancer center and the rehabilitation experts who manage lymphedema clinics. Hopefully, this procedure will diminish referrals to the clinics but some patients will still need supportive care. Additionally, an unbiased limb assessor who can measure limb volume and objectively determine the presence of lymphedema takes their time and offers dedicated interest. Affiliation with a research group and a cancer center is important to assess the results of prospective cohorts.

## 3. Postoperative Follow-Up

### 3.1. Measuring Lymphedema

A rigorous follow-up of patients with a pre-specified protocol and tool to measure lymphedema is important. There are many different definitions of lymphedema and perhaps even more tools to measure it [40–45]. A difference in circumference of 2 centimeters or more between matched points on the two arms or a volume difference of 10% are some of the various ways to define lymphedema [40,41]. Tools can include questionnaires [46–48], limb measurements, bioimpedance spectroscopy and volumetric measurements such as perometry [43].

In our institution, both limb measurements and patient-reported outcomes are collected. Preoperative measurements are ideally obtained and limb changes over time are documented. Contralateral limb measurements are useful as well, but may differ based on hand dominance (Figure 4). Measurements are repeated every 6 months, or sooner if the patient feels they are swelling. Body mass index is tracked as limb volume is sensitive to weight changes. Circumferential limb measurements are performed every 4 cm between 0 (wrist) and 40 cm (upper arm). The volume of each limb is calculated using a cone calculation formula, assimilating each 4 cm segment of the arm to a truncated cone. Regarding patient-related outcomes, the LYMQOL questionnaire is favoured for ease of use, with a lower respondent burden: 21 questions cover four relevant domains (function, appearance, symptoms and mood) [48].

Our current immediate lymphatic reconstruction experience includes 52 patients, in which an average of two (range one–five) lymphovascular anastomoses were performed. No procedures were aborted and no patient had a complication requiring a return to the operating room. The number of patients who had an ILR for cutaneous malignancy was 30, 19 had it for breast cancer and 3 had it for colorectal, ovarian or penile malignancies. A total of 40 patients had an upper limb ILR. With respect to the upper limb, with an average of approximately 18 months follow-up, one patient developed lymphedema.

Compression stockings and lymphatic massage are not routinely used postoperatively. Patients who develop lymphedema are referred to the lymphedema clinic. After assessment, they are prescribed dressings, stockings and therapy as appropriate. At the clinic, they are treated similarly to patients who have not had an ILR, meaning there are no contraindications for them to undergo usual lymphedema therapy.

## 4. Education and Research

### 4.1. Establishing a Research Setting

An ILR program should ideally take place in a research setting. This includes the involvement of an institutional review board (IRB), as well as a research team including a research coordinator, nurses, trainees and other valuable team members. Result assessment can be time consuming but it is necessary to monitor the effectiveness of the program. It can take the form of a prospective trial or randomized controlled trial, with the latter being for as long as there is equipoise. Ongoing research is stimulating and nudges the program forward by continuing to ask questions and aiming for improvement. The interest in ILR lies only partly in the technical challenge; it equally resides in academic inquisition and curiosity.

### 4.2. Development of an Educational Curriculum

To grow the program further, one can consider the development of an educational curriculum. This can start in a simulation lab to train surgical teams (Figure 5). Oncologic and plastic surgeons assisting one another under the microscope takes practice. Supermicrosurgery requires additional skill development for the microsurgeon. Non-living animal models are available for this purpose. Top-quality microinstruments are required for the smallest calibre anastomotic practice.

While technical teaching is a major focus of our educational curriculum, non-technical skills are incorporated as well. The team dynamics, both between the two surgeons and between the surgeons and nurses, need nurturing. Often, teams have previously gelled from

working together on breast reconstruction or other oncologic reconstructive cases. What is unique here, however, is the microsurgeon being more hands-on during the resection, by dissecting lymphatics for length toward tumour-bearing nodes, and the surgical oncologist being attentive to the need for length on lymphatic and recipient vessels, such as dissecting and preserving veins that might otherwise be clipped. The communication and operative flow can be simulated in the lab, and also be practiced in the workplace environment with debriefing afterward. The Non-Technical Skills for Surgeons (NOTSS) system is particularly useful here. NOTSS provides a vocabulary around four categories of skill: Situation awareness includes gathering and understanding information, as well as projection and anticipation of the future. *Decision-making* involves considering options, selecting and communicating options and implementing and reviewing decisions. Communication and teamwork comprises exchanging information, establishing a shared understanding and coordinating team activities. Finally, leadership encompasses setting and maintaining standards, supporting others and coping with pressure [49]. Any identified gaps in non-technical skills can be taken back to the simulation lab to hone.

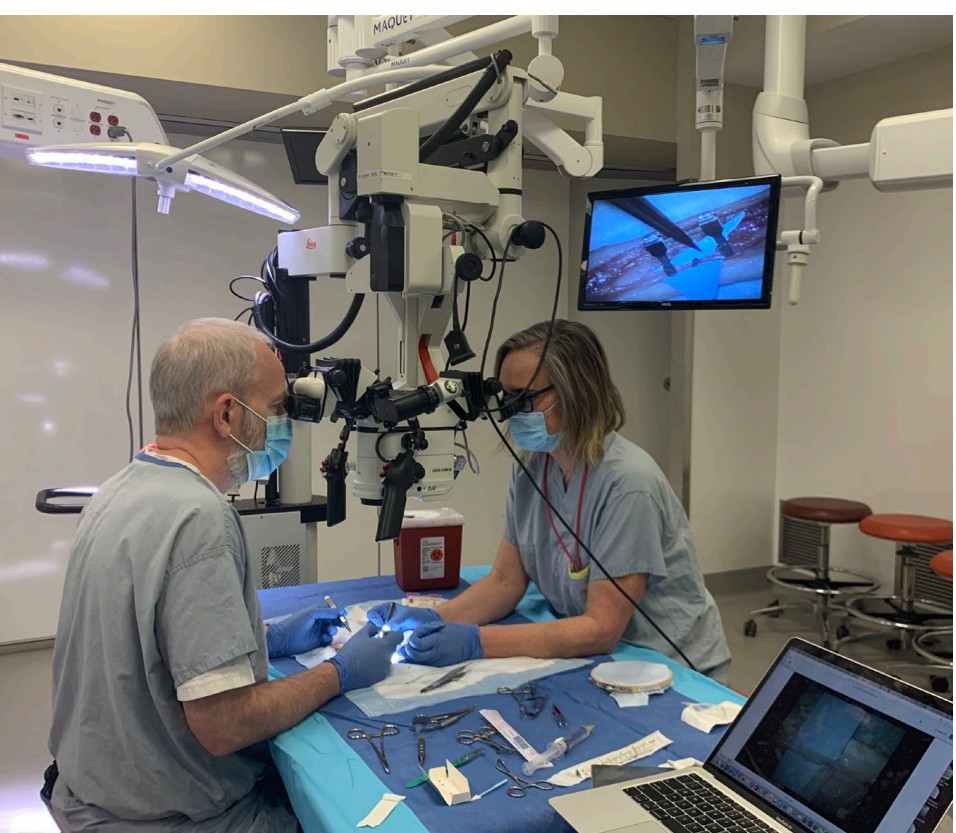

**Figure 5.** Oncologic and plastic surgeons train together in the ATSSL simulation lab in Calgary.

## 5. Conclusions

In summary, ILR is an exciting approach to prevent lymphedema in patients with breast cancer. Figure 6 summarizes the steps required to elaborate an ILR program in the form of a flowchart. In the context of improving survivorship, it is our challenge and responsibility to mitigate the burden of lymphedema on the quality of life of our patients. While ILR may be considered resource-intensive at first, it is an investment that we anticipate to be well-returned in terms of cost, teamwork dynamics, academic research and, above all, improved quality of life of our patients.

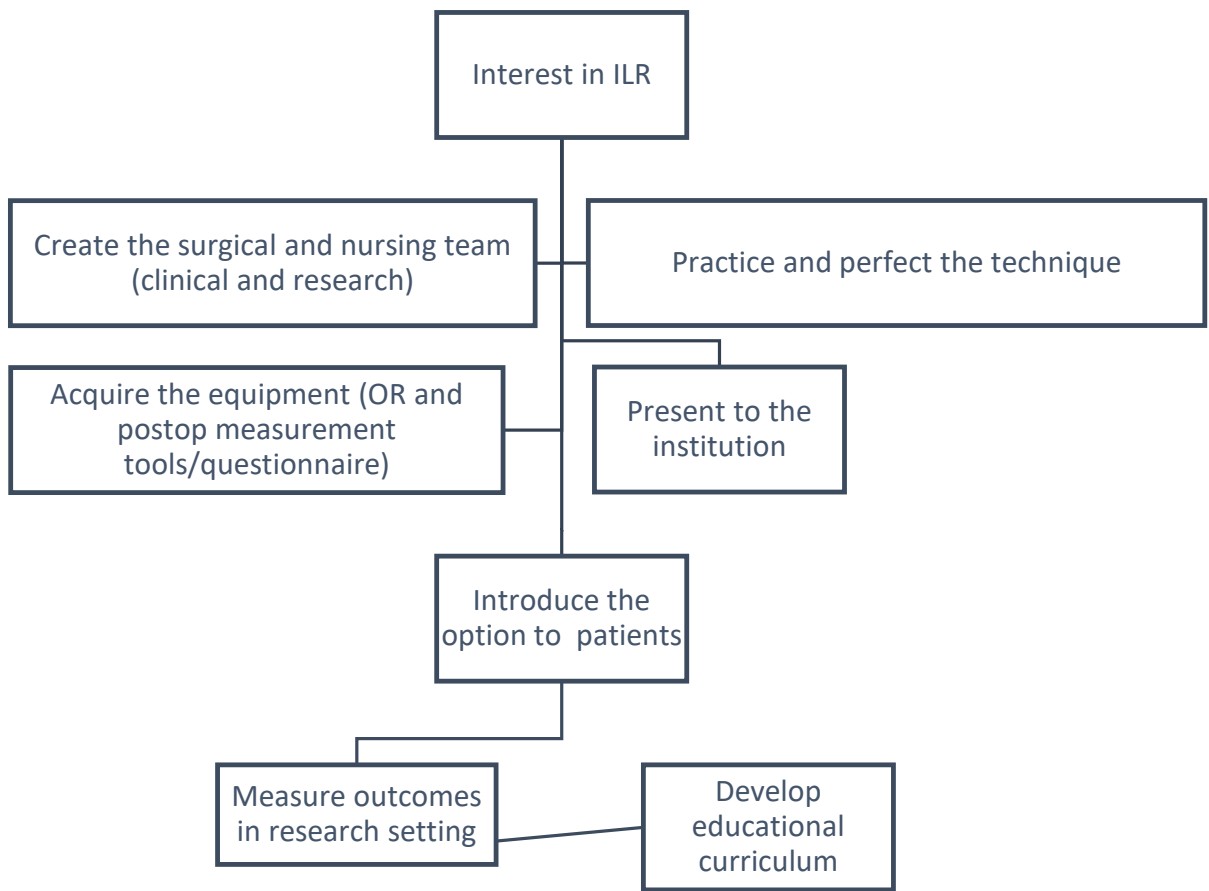

**Figure 6.** Summary of possible steps in elaborating an ILR program.

**Author Contributions:** Conceptualization, M.D. and C.T.-O.; methodology, M.D., J.G.M. and C.T.-O.; writing—original draft preparation, M.D., J.G.M. and C.T.-O.; writing—review and editing, M.D., J.G.M. and C.T.-O.; supervision, C.T.-O. All authors have read and agreed to the published version of the manuscript.

**Funding:** This research received no external funding.

**Informed Consent Statement:** Written informed consent has been obtained from the patient(s) to publish this paper.

**Data Availability Statement:** The data presented in this study are available on request from the corresponding author.

**Conflicts of Interest:** The authors declare no conflict of interest.

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
