# Peer review of "Mitigating Breast-Cancer-Related Lymphedema—A Calgary Program for Immediate Lymphatic Reconstruction (ILR)"

_curroncol, doi:10.3390/curroncol30020119_

Round 1
Reviewer 1 Report
dear author team
thank you for submitting your enthusiastic paper to our journal.
LVA surgery is starting to get to grip in clinics globally. Many hours of webinars are focussing on how to get a pogram started. I feel your paper dresses all interesting subjects in an evidence based and practical manner.
For this I feel your paper will be of great interest to the reader.
perhaps I might suggest to also share with us a protocol with a stepwise description of what happens when;
indication for immediate lump recon during MDT meeting;
preop measurements; whenwhat, preop therapy?
operative is described
postoperative; questionnaires and tests, physio/lymph drainage/pressure dressings yes/no etc.
this will be of great help for the starters.
thank you
Author Response
Thank you for your thorough review and insightful comments. Your interest for this subject and paper are appreciated.
Based on your suggestion, we have included a flowchart that summarizes the steps to elaborate an ILR program in section 5. The indications for ILR have been added in section 2.2. Timing of preoperative measurements and therapy have been further elaborated in section 2.2. Postoperative questionnaires and tools are detailled in section 3.1.
We thank you again and hope this will address the issues raised during the review.
Best regards,
Melina Deban
Reviewer 2 Report
This study describes the introduction of a new surgical technique: Immediate Lymphatic Reconstruction (ILR), and describes briefly the outcomes of a cohort of 52 patients undergoing the procedure as part of primary therapy for cancer
Hypothesis - nil stated
Methods:
- inclusion/exclusion criteria are not stated ( eg previous surgery, axillary irradiation)
- "instrument of measurement" of lymphoedema appears to be a bespoke tool - this could do with more description if the results are to be compared
-the study is strong on the anastomotic details
Results:
- these are more narrative, relating to the introduction of a new technique
- were any "proctoring protocols" followed?
- did the authors devise an educational curriculum?
- how was this assessed?
- were there any technical failures?
- post op scintigraphy?
- returns to theatre?
- were there historical controls?
Conclusions
- appropriate
Illustrations
- good
Tables:
the results (of the 52 patients) could be summarized in a table
the "introduction" of a curriculum could be shown in a flow diagram
Author Response
Thank you for your thorough review and insightful comments.
Based on your suggestion, the indications for ILR have been added in section 2.2.
The source of the instrument of measurement of lymphedema has been cited with an accessible source.
The aim of this paper is indeed a narrative description of the steps required to elaborate an ILR program. The data for the first 52 patients is not fully mature at this point and we hope to be able to report our early experience in the near future in a rigorous way.
Educational curriculum and technical failure have been further elaborated in the text.
We have included a flowchart that summarizes the steps to elaborate an ILR program in section 5.
We thank you again and hope this will address the issues raised during the review.
Best regards,
Melina Deban
Round 2
Reviewer 2 Report
I accept the changes made by the authors